# Towards a webcam-based snow cover monitoring network: methodology and evaluation

Céline Portenier[1], Fabia Hüsler[2], Stefan Härer[3], and Stefan Wunderle[1]

[1]Institute of Geography and Oeschger Centre for Climate Change Research, University of Bern, Bern, Switzerland
[2]Federal Office for the Environment FOEN, Ittigen, Switzerland
[3]Professorship Ecoclimatology, Technical University of Munich, Freising, Germany

**Correspondence:** Céline Portenier (celine.portenier@giub.unibe.ch)

**Abstract.** Snow cover variability has a significant impact on climate and environment and is of great socio-economic importance for the European Alps. Terrestrial photography offers a high potential to monitor snow cover variability, but its application is often limited to small catchment scales. Here, we present a semi-automatic procedure to derive snow cover maps from publicly available webcam images in the Swiss Alps and propose a procedure for the georectification and snow classification of such images. In order to avoid the effort of manually setting ground control points (GCPs) for each webcam, we implement a novel registration approach that automatically resolves camera parameters (camera orientation, principal point, field of view (FOV)) by using an estimate of the webcams' positions and a high-resolution digital elevation model (DEM). Furthermore, we propose an automatic image-to-image alignment to correct small changes in camera orientation and compare and analyze two recent snow classification methods. The resulting snow cover maps indicate whether a DEM grid is snow-covered, snow-free, or not visible from webcams' positions. GCPs are used to evaluate our novel automatic image registration approach. The evaluation reveals in a root mean square error (RMSE) of 14.1 m for standard lens webcams (FOV $< 48°$) and a RMSE of 36.3 m for wide-angle lens webcams (FOV $\geq 48°$). In addition, we discuss projection uncertainties caused by the mapping of low resolution webcam images onto the high-resolution DEM. Overall, our results highlight the potential of our method to build up a webcam-based snow cover monitoring network.

## 1 Introduction

Snow is an essential natural resource. Because snow has a much higher albedo compared to other natural land surfaces, its areal extent plays an important role in the Earth's energy balance. In alpine regions, snow plays a key role in the hydrologic cycle. It acts as water storage and accounts for a substantial portion of the total runoff. Information about spatial and temporal snow distribution is therefore essential for monitoring water resources and predicting runoff (Jonas et al., 2009), and it is of crucial importance for water supply and hydropower production. In addition, seasonal snow cover not only plays an important role for the development of ecosystems but has a high economic value for winter tourism as well.

Most commonly used methods to monitor snow cover variability are based on in situ measurements and satellite remote sensing. In situ measurements, e.g., from ground-based monitoring networks, provide accurate and long time series of local snow sites and can be used, for example, for long-term trend analyses (e.g., Laternser and Schneebeli, 2003; Marty, 2008;

Klein et al., 2016). These measurements, however, might not capture the spatial variability of snow cover. In contrast to in situ measurements, remote sensing data can provide spatially comprehensive information on snow cover extent. In particular, optical remote sensing is widely used to study snow cover variability (e.g., Foppa and Seiz, 2012; Hüsler et al., 2012; Metsämäki et al., 2012; Wunderle et al., 2016). The main limiting factor of optical remote sensing techniques is cloud coverage.

According to Dumont and Gascoin (2016), the yearly average of pixels hidden by clouds is about 50% in the Pyrenees and 60% in the Austrian Alps. In addition, large uncertainties exist in shadowed or forested areas. Moreover, the sensor resolution (e.g., 250 m or 1.1 km resolution of the MODIS and AVHRR sensor respectively) may limit the capture of small-scale variability of snow cover, especially in complex, mountainous terrain. The emergence of new techniques based on airborne digital photogrammetry and terrestrial photography enables to extract snow cover information with high spatial and temporal resolutions.

Unmanned areal systems (UAS) enable the generation of high-resolution digital surface models that can be used to map the small scale variability of snow depth (e.g., Bühler et al., 2016; De Michele et al., 2016). However, UAS are often associated with high costs and its spatial coverage and temporal resolution is limited. In addition, weather constraints due to strong winds or precipitation can restrict the use of UAS, especially at high elevations.

In this work, we suggest the use of publicly available webcam images and present a semi-automatic procedure to generate

snow cover maps from such images. This work builds on and extends the Master's thesis by Dizerens (2015). We focus on the Swiss Alps, where several thousands of public outdoor webcams are readily available online, resulting in a relatively dense sampling to study snow cover variability over a large area. Webcams are a cost-effective and efficient way to monitor snow cover variability in mountainous regions at high spatio-temporal scales. Most webcams offer images within a one-hourly to 10-minute interval. The spatial resolution depends on the image resolution, a webcam's field of view (FOV), the distance of

the terrain to the webcam, as well as the slope and orientation of the terrain (see Sect. 5 for an in-depth discussion). Webcams may offer detailed analyses of snow cover on steep slopes due to their oblique view on the mountains. Moreover, webcams can provide snow cover information even under cloudy weather conditions as long as cloud cover and fog do not disturb the view on the ground. Therefore, webcams offer an unique potential to complement and evaluate satellite-derived snow information. For instance, Piazzi et al. (2019) have shown that webcam images can be leveraged to assess the consistency of Sentinel-2 snow

cover information. However, the areal coverage of webcam-based snow cover information depends on the number of cameras used, their FOV, and their positioning in the field. In addition, public webcams provide images in the visible spectrum only and with varying image quality, which makes an accurate classification of snow cover challenging.

Terrestrial photography is an increasingly used observation method in different research disciplines such as glaciology (e.g., Corripio, 2004; Dumont et al., 2011; Huss et al., 2013; Messerli and Grinsted, 2015) and snow cover studies (e.g., Schmidt

et al., 2009; Farinotti et al., 2010; Härer et al., 2013; Pimentel et al., 2014; Härer et al., 2016; Liu et al., 2015; Fedorov et al., 2016; Revuelto et al., 2016; Arslan et al., 2017; Millet et al., 2018). However, most of these studies use single cameras and thus are limited in areal coverage. In particular, they either require known camera parameters (i.e., extrinsic and intrinsic camera parameters such as the camera orientation or the FOV of the camera) or require significant manual user input (e.g., ground control points (GCPs)) to georectify terrestrial photography. Since camera parameters are not readily available for

public webcams, and manually setting GCPs for a large number of cameras is time-consuming, these methods are of limited

application for our purposes. Therefore, we implement a processing scheme that minimizes manual user input by automation. Our georectification approach registers a webcam image with a digital elevation model (DEM). This image-to-DEM registration automatically resolves the required webcam parameters, such as the camera's orientation and its FOV by using an estimate of the webcam's position only.

In literature, many different snow classification techniques exist to detect snow cover in terrestrial camera images. Some studies determine the snow covered area using manual interpretation (Farinotti et al., 2010; Liu et al., 2015) or by manually selecting appropriate threshold values for each single image (Schmidt et al., 2009) or for a set of images (Floyd and Weiler, 2008). On the other hand, many automatic approaches exists as well, such as methods applying image clustering techniques (Pimentel et al., 2014; Millet et al., 2018; Rüfenacht et al., 2014), other statistical methods (e.g. Salvatori et al., 2011; Härer

et al., 2016), or using supervised learning classifiers (Fedorov et al., 2016) to distinguish snow from snow-free areas. The main challenge of these methods is to detect snow cover in shadowing areas or to differentiate between dark, shadowed snow pixel and other canopy pixels such as bright rock surfaces (Rüfenacht et al., 2014; Härer et al., 2016; Arslan et al., 2017; Manninen and Jääskeläinen, 2018). The study of Härer et al. (2016) takles the problem of undetected snow cover in shadowing regions. Härer et al. (2016) propose to apply the blue band classification by Salvatori et al. (2011) and subsequently use

principal component analysis (PCA) to separate shaded snow cover from sunlit rock surfaces. Recently, Fedorov et al. (2016) propose to train machine learning models to classify snow cover in terrestrial camera images. While Fedorov et al. (2016) report superior performance to handcrafted methods on data that is sufficiently similar to the training data, such models do not generalize well to data that deviates significantly from the training data. Moreover, acquiring data suitable for training such models is expensive, since it requires to label every single pixel in a set of training images by hand. In this study, we test and

compare the snow classification approaches proposed by Salvatori et al. (2011) and Härer et al. (2016) within our framework. Combined with an automatic image-to-image alignment to correct small changes in the camera orientation, our procedure can be applied to webcam images to generate snow cover maps with a minimal effort. To assess the accuracy of our automatic snow cover mapping, we analyze and evaluate the components of the processing chain with a focus on automatic image-to-DEM registration.

This work is organized as follows: in Sect. 2, the webcam data, DEM, and orthophoto used in this work are described. In Sect. 3, we present the proposed methods of our procedure. Qualitative examples of snow cover maps and a comparison of the applied snow classification methods are shown in Sect. 4, followed by a detailed evaluation of the mapping accuracy in Sect. 5. Finally, we discuss the advantages and limitations of our procedure (Sect. 6), before concluding in Sect. 7.

## 2   Data

### 2.1   Webcam images

The website www.kaikowetter.ch offers a network of about 520 outdoor webcams observing the current snow conditions in and around Switzerland. Most of these webcams were installed by mountain railway operators, restaurants, hotels, and private citizens. Since November 2011, we are archiving one image per day of each webcam from this website and extend our archive

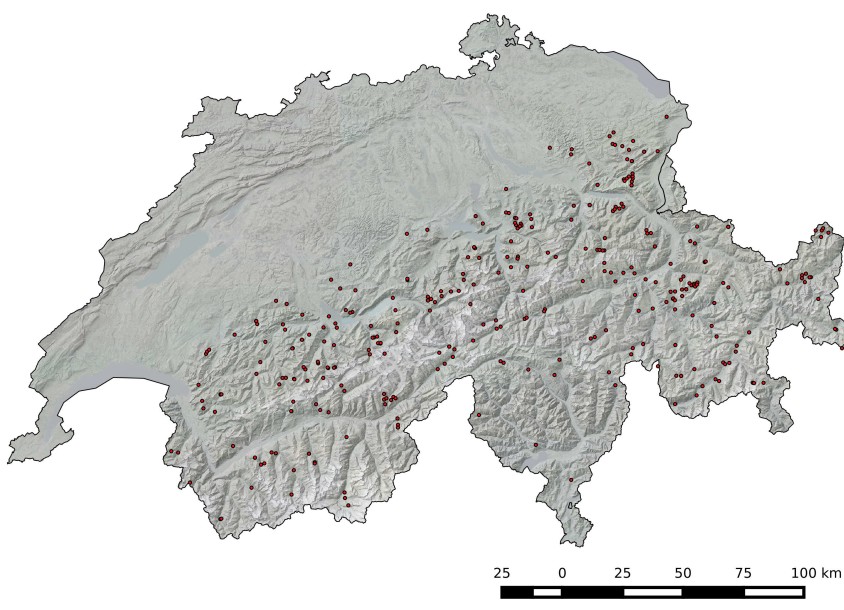

**Figure 1.** Locations of 297 webcams (red points) in the Swiss Alps. Background data: SWISSIMAGE and swissALTI3D, source: Swiss Federal Office of Topography.

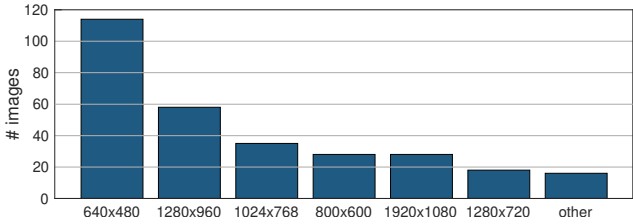

**Figure 2.** Image pixel resolution of the selected webcams.

continuously with webcam images from other web-sources. To apply our procedure to a given webcam, two requirements have to be fulfilled: the mountain silhouette has to be visible on the webcam image, i.e., it is not obscured by trees or buildings. About 70% of all the webcams provided by kaikowetter.ch satisfy this requirement. The other approximately 30% can not be used due to obstacles between silhouette and webcams or since no mountain silhouette is visible at all. In addition, the location of a webcam has to be known. Up to now, we have manually estimated the locations of 297 webcams in the Swiss Alps (see Fig. 1). They are located at elevations ranging from 800 m to 3900 m a.s.l.. The pixel resolution of these webcam images ranges from $640{\times}480$ up to $1920{\times}1080$ pixels (see Fig. 2).

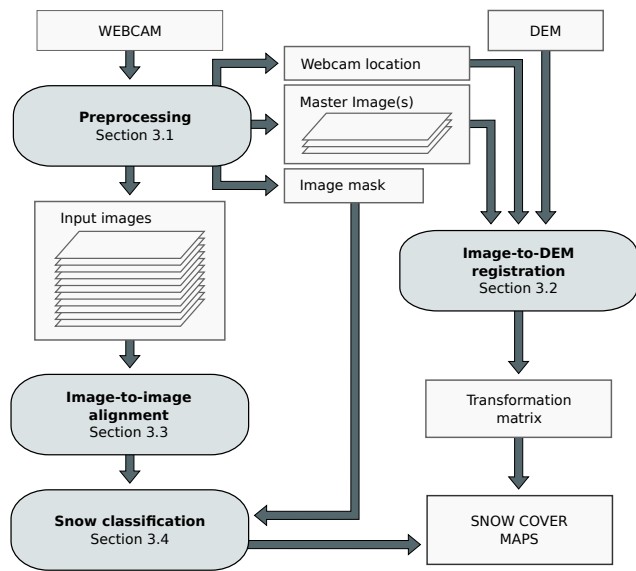

**Figure 3.** Overview of the proposed procedure. It consists of four major steps: preprocessing, automatic image-to-DEM registration, automatic image-to-image alignment, and automatic snow classification. Image-to-DEM registration results in a transformation matrix that is used to project the snow-classified pixels onto a map.

## 2.2 Swiss geodata

We use the swissALTI$^{3D}$ DEM and the orthophoto SWISSIMAGE, produced by the Swiss Federal Office of Topography (swisstopo, 2013a, b). The DEM covers Switzerland and Liechtenstein and has a spatial resolution of 2 m. It was created using airborne laser scanning data (below 2000 m a.s.l.) or stereocorrelation of areal photographs (above 2000 m a.s.l.) and features
5  an accuracy of 0.5 m and 1 to 3 m on average, respectively. The orthophoto SWISSIMAGE is composed of digital aerial orthophotographs of Switzerland, featuring a spatial resolution of 0.1 m in the Swiss Lowlands and 0.25 m in the Swiss Alps.

## 3 Methods

The proposed procedure consists of four major steps: preprocessing, automatic image-to-DEM registration, automatic image-to-image alignment, and automatic snow classification (see Fig. 3 for an overview). In the preprocessing step (Sect. 3.1),
10  manual user input is required to estimate the webcam's location, to select a representative image for image-to-DEM registration (hereafter referred as Master Image), and to provide an image mask. Second, the selected Master Image is automatically registered with the DEM to derive the unknown camera parameters, such as orientation and FOV of the webcam (Sect. 3.2). Successful image-to-DEM registration results in a transformation matrix that relates each pixel of the Master Image to its 3D coordinates. Since an image series of a webcam is usually not perfectly aligned, we automatically align images to the selected

Master Image (Sect. 3.3). This enables the use of the same transformation matrix for all webcam images. Finally, each image is automatically snow-classified (Sect. 3.4). Using the transformation matrix, a georeferenced snow cover map can be generated.

## 3.1 Preprocessing

First, a webcam's location and its installation height above ground is estimated manually. This is achieved by considering the position of objects visible in the webcam image, the orthophoto SWISSIMAGE, and additional information provided by the webcam owner (e.g. the name of a restaurant where the webcam is mounted). In some cases, touristic photographs and images from Google Street View help to improve the location estimation. As mentioned in Sect. 2, we have estimated the locations of 297 webcams (see Fig. 1). Next, at least one Master Image per webcam is selected. This image has to be representative for all other images of the same webcam, and should feature high contrast between the mountains and the sky for automatic image-to-DEM registration. Under clear sky conditions, most webcam images are suited to serve as Master Image. Finally, a so-called input mask can be prepared to define image regions that should be ignored in the snow map generation procedure. Such regions can be trees, buildings, or other fixed infrastructure, and are defined on the Master Image.

## 3.2 Automatic image-to-DEM registration

The registration of an image with a DEM requires a common feature space. As in the study of Baboud et al. (2011) and Fedorov et al. (2016), we make use of mountain silhouettes, which are among the most salient structural features in mountainous natural environments. Gaussian filtering and Sobel edge detection are applied to the Master Image to reduce noise and extract the structural features from the images. Next, the mountain silhouette is automatically detected from the edge image (see Fig. 4). Our silhouette extraction is based on the assumption that the mountain silhouette is the uppermost edge line that spans the full width of the image. It starts at the top left pixel in the edge image and looks for the first edge pixel in the first column. Once a pixel is found, the algorithm iteratively searches in a $7 \times 7$ pixel neighborhood for other edge pixels until a continuous line is found that spans the full width of the image. If no such edge line is found, the algorithm starts again at the next edge pixel in the first column of the image.

To derive the unknown camera parameters, the extracted mountain silhouette is registered with mountain silhouettes extracted from virtually rendered DEM images. These DEM images are generated by projecting the DEM point cloud from its world coordinate system via a camera coordinate system to an image coordinate system (see Fig. 5 and 6) by using a pinhole camera model. To reduce the computational complexity, only DEM points that are visible from the point of view of the webcam are considered. For this purpose, the viewshed generation module of the Photo Rectification And ClassificaTIon SoftwarE (PRACTISE V.1.0; Härer et al., 2013) is used to generate a $360°$ visibility map from the point of view of the webcam. The projected DEM points $\mathbf{p}'$ of the virtual DEM image are computed by multiplying the visible DEM points $\mathbf{p}$ by the inverse of a camera matrix $\mathbf{C}$, a perspective projection matrix $\mathbf{P}$, and a viewport matrix $\mathbf{D}$:

$$\mathbf{p}' = \mathbf{D}\mathbf{P}\mathbf{C}^{-1}\mathbf{p}. \tag{1}$$

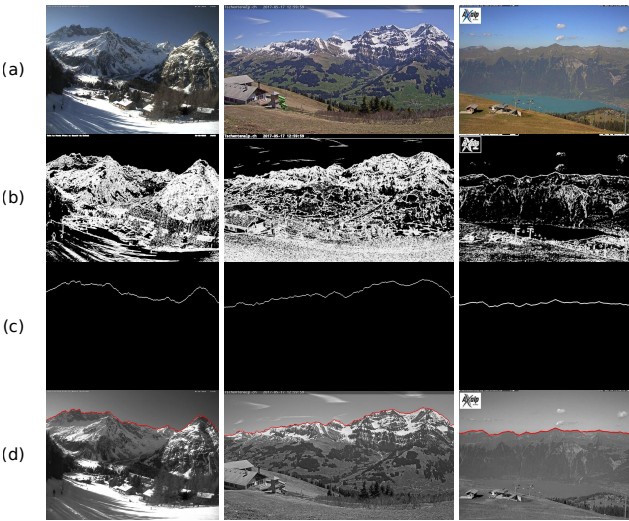

**Figure 4.** Three examples of automatic silhouette extraction. (a) Webcam images, (b) extracted edges using Sobel edge detection, (c) detected mountain silhouettes, and (d) mountain silhouettes (red) superimposed on grayscale webcam images.

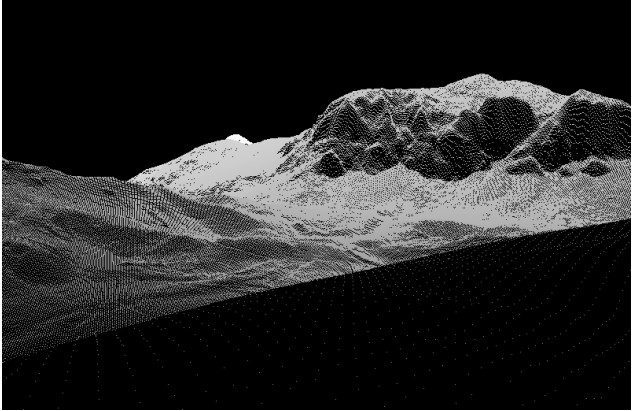

**Figure 5.** Sample rendering of a digital elevation model (DEM) using a pinhole camera model.

The camera matrix $\mathbf{C}$ transforms from camera coordinates to world coordinates and is defined by extrinsic camera parameters, i.e., the camera's location and orientation with respect to the known world reference frame. It is given by

$$\mathbf{C} = \begin{bmatrix} \mathbf{x_c} & \mathbf{y_c} & \mathbf{z_c} & \mathbf{cop} \\ 0 & 0 & 0 & 1 \end{bmatrix}, \tag{2}$$

where $\mathbf{cop}$ is the camera's location and $\mathbf{x_c}$, $\mathbf{y_c}$, and $\mathbf{z_c}$ are the three vectors of the camera coordinate system that define its orientation, i.e., the roll, pitch, and yaw angle. The perspective projection matrix $\mathbf{P}$ transforms objects into canonic view

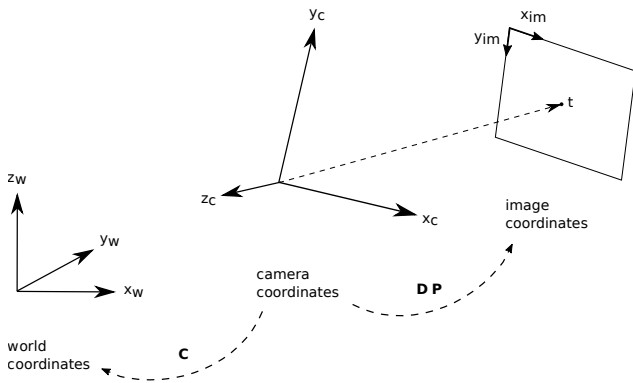

**Figure 6.** World, camera, and image coordinate systems and its transformations using camera matrix $\mathbf{C}$, perspective projection matrix $\mathbf{P}$, and viewport matrix $\mathbf{D}$.

volume (i.e. a cube) so that the image points are normalized view coordinates in the range $[-1,1] \times [-1,1] \times [-1,1]$. It is defined by intrinsic camera parameters and is given by

$$
\mathbf{P} = \begin{bmatrix} \frac{1}{a \cdot \tan(FOV/2)} & 0 & 0 & 0 \\ 0 & \frac{1}{\tan(FOV/2)} & 0 & 0 \\ 0 & 0 & \frac{near+far}{near-far} & \frac{2 \cdot near \cdot far}{near-far} \\ 0 & 0 & -1 & 0 \end{bmatrix},
\tag{3}
$$

where $a$ is the image aspect ratio and $near$ and $far$ are the distances to a near and a far plane that limit the infinite viewing
5    volume. To finally transform to pixel coordinates $(x_{im}, y_{im}) \in [x_o...x_1] \times [y_0...y_1]$, the viewport matrix, given by

$$
\mathbf{D} = \begin{bmatrix} (x_1 - x_0)/2 & 0 & 0 & (x_0 - x_1)/2 \\ 0 & (y_1 - y_0)/2 & 0 & (y_0 - y_1)/2 \\ 0 & 0 & 1/2 & 1/2 \\ 0 & 0 & 0 & 1 \end{bmatrix}
\tag{4}
$$

is applied. It scales the projected pixels to a certain image size and translates them such that the origin of the image coordinate system is at the upper left corner. Since we use homogeneous coordinates, we apply perspective division to obtain pixel coordinates. Using this camera model, virtual DEM images can be generated by sampling the unknown parameters (i.e., the
10    three orientation vectors $\mathbf{x_c}$, $\mathbf{y_c}$, and $\mathbf{z_c}$ of the camera and the FOV).

     To estimate the ground truth camera parameters, we propose a silhouette matching procedure. Similar to before, the mountain silhouettes are extracted from the rendered DEM images using the method described above. Given two silhouettes, i.e., the

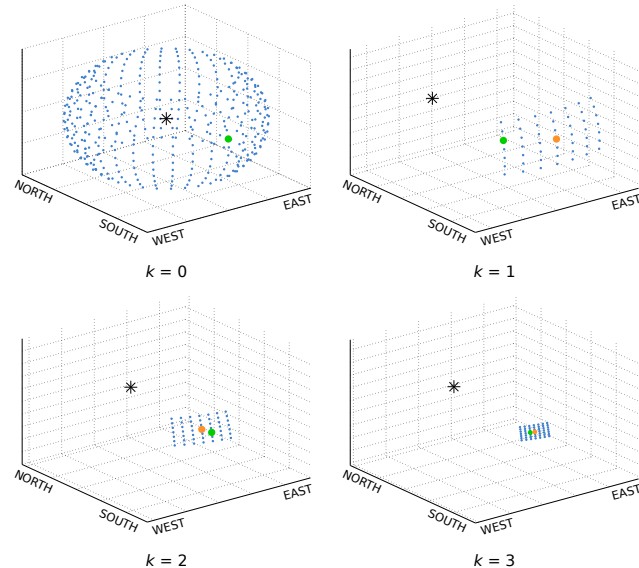

**Figure 7.** Viewing directions (blue points) of a camera (asterisk) during image-to-DEM registration. The green dots indicate the viewing directions with the best score, and the orange dots indicate the best viewing directions of the previous scale. An example is shown for vertical and horizontal rotations from scale $k = 0$ to $k = 3$.

Master Image silhouette and a silhouette extracted from a sampled DEM rendering, we define a score function based on 2D cross-correlation to quantify how well the two silhouettes match:

$$score = \alpha \cdot w_1 + (1 - \alpha) \cdot w_2 \,, \tag{5}$$

where $w_1$ is the normalized maximum response of cross-correlation, and $w_2$ is the normalized image space offset defined by the distance between the pixel location of the maximum response and the image center. The final score is the weighted sum using a user-defined parameter $\alpha$. To estimate the camera parameters, we seek for the parameters that maximize this score.

To efficiently search for the best matching silhouette pair, silhouette matching is performed on multiple scales $k$. On each scale, the algorithm rotates the camera coordinate system horizontally and vertically (see Fig. 7) and searches for the highest score. On scale $i$, the estimated parameters of scale $i - 1$ are used as initialization and the camera coordinate system is rotated $n_x$-times around the z-coordinate of the world coordinate system and $n_y$-times around the x-axis of the camera coordinate system. On scale $k = 0$, the parameters are initialized randomly. The horizontal and vertical rotation steps are called strides $s_x$ and $s_y$ respectively. On scale $k = 0$, we set an initial stride of $s_x = 360°/n_x$ (with $n_x = 20$) and $s_y = 90°/n_y$ (with $n_y = 12$). For all scales $k > 0$, the horizontal and vertical strides are recursively defined by

$$s_{x_i} = \frac{3 s_{x_{i-1}}}{n_x} \text{ and } s_{y_i} = \frac{3 s_{y_{i-1}}}{n_y} \,. \tag{6}$$

To approximate the roll angle of the camera, we additionally rotate the x-coordinate of the camera matrix on each scale $m = 5$ times around the viewing direction once the image space offset $w_2$ is smaller than 10 pixels. An initial stride of $s_m = 3°/m$ is set and decreased each scale by

$$s_{m_k} = \frac{3 s_{m_{k-1}}}{m}.$$

(7)

5     Instead of estimating the FOV manually, our procedure can also optimize the FOV of the webcam by first iterating the horizontal FOV of $30°$ by $5°$ to a FOV of $90°$ in scale $k = 0$. The best matching silhouette pair defines the initial FOV estimate. Once the image space offset $w_2$ is smaller than 20 pixels, the FOV can be estimated more accurately by evaluating different FOVs at each iteration: the FOV is iterated at each viewing direction $f = 5$ times around the initial FOV with an initial stride $s_{FOV} = 2°$, decreasing each scale by

10     $s_{f_k} = \frac{3 s_{f_{k-1}}}{f}.$

(8)

The weighting parameter $\alpha$ (Eq. 5) is a function of scale $k$. On scale $k = 0$, we set $\alpha = 1$, such that the final score is mainly determined by the maximum response of cross-correlation $w_1$. The normalized image space offset $w_2$ is ignored, since it would mainly correspond to an offset of a wrongly matched silhouette pair. $w_2$ becomes important for scales $k > 0$, once the viewing direction estimate is reasonably accurate. The smaller the distance of the maximum response to the image center, the better the two silhouettes match. Therefore, $\alpha$ is set to a low value ($0.1$). Once the roll angle and FOV is resolved, both measures, $w_1$ and $w_2$, are set equally ($\alpha = 0.5$), since both the smallest offset and the highest response value have to be estimated.

To find the best score efficiently, the virtual DEM images are rendered with a lower resolution in the first scales. Starting with a width of $w = w_{orig}/8$ and height of $h = h_{orig}/8$ in scale $k = 0$, the width and height are doubled until the original image size is reached in scale $k = 3$. Experiments have shown that image-to-DEM registration requires around 12 scales until the best matching silhouette pair with an image space offset of 0 is found. This best matching silhouette pair results in a transformation matrix that relates each pixel of the Master Image to its real 3D coordinates.

### 3.3    Automatic image-to-image alignment

Most webcams are exposed to wind that may lead to small changes in the camera orientation. Moreover, for a few webcams major variations in orientation can occur due to human interaction, intentionally or unintentionally. While small orientation changes may occur every day, we observe major camera movements rarely, at most monthly. Because image-to-DEM registration is computationally expensive and mountain silhouettes cannot be detected on each webcam image due to cloud cover or low contrast conditions, each webcam image is automatically aligned to its Master Image by solving for a homography $\mathbf{H}$. A homography is a projective transformation between two images with the same camera position but different orientation and is used to relate the two images so that they can be aligned.

30     We use the Scale Invariant Feature Transform (SIFT; Lowe, 2004) to detect structural features in a webcam image and its corresponding Master Image. It transforms an image into a collection of local feature vectors that consist of a SIFT keypoint (image location) and a SIFT descriptor that is highly distinctive and invariant to illumination, position, and scale. After the

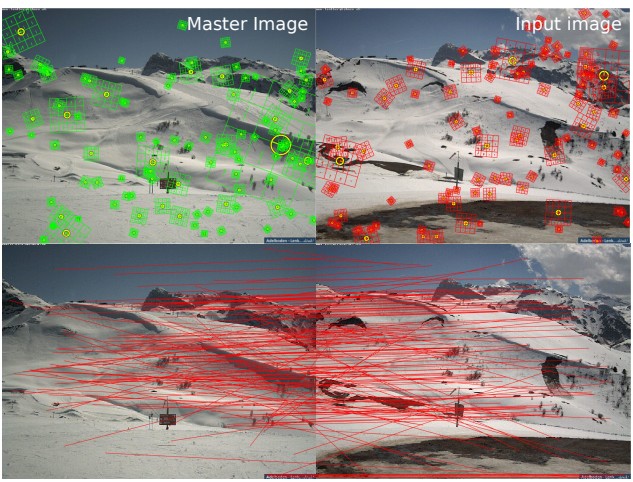

**Figure 8.** SIFT features of a Master Image and an input image and corresponding matches between all features. To simplify the illustration, we show a subset of 100 randomly selected SIFT features per image.

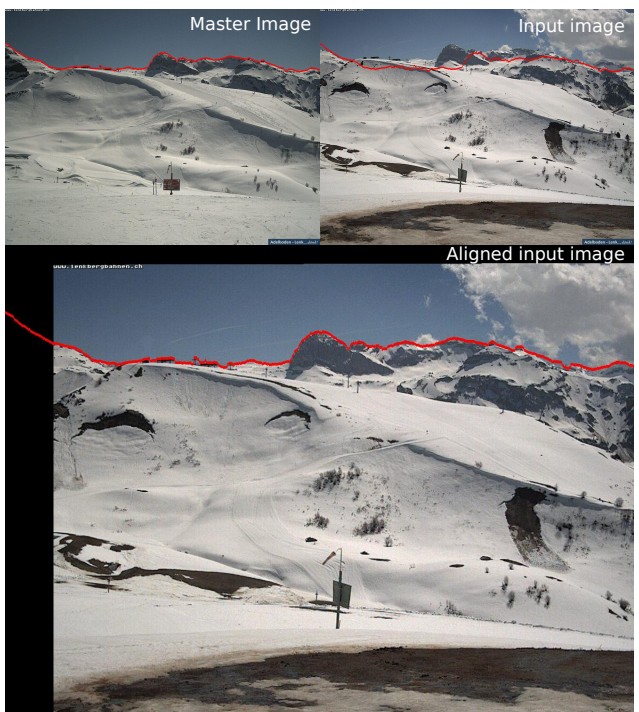

**Figure 9.** Example of an arbitrary input image that is aligned to a corresponding Master Image. The mountain silhouette extracted from the Master Image is shown in red.

feature detection, the features are matched across the two images (see Fig. 8). The similarity between two feature vectors is given by their Euclidean distance. Since the number of potential matching features can be quite large, we approximate this distance using an algorithm called Best-Bin-First (see Lowe, 2004). We use the SIFT implementation from the open source library VLFeat (Vedaldi and Fulkerson, 2010).

5    A homography $\mathbf{H}$ is a $3 \times 3$ matrix. Since scale is arbitrary, $\mathbf{H}$ has eight unknown parameters. Therefore, at least four point correspondences (x/y image coordinates) are needed to solve for $\mathbf{H}$. Since not all matched pairs are correct, the homography is estimated using the best matching feature points. For this purpose, we use the robust fitting model RANdom SAmple Consensus (RANSAC; Fischler and Bolles, 1981). RANSAC randomly selects four pairs of corresponding points to calculate the homography, transforms all points from one image to the other using the found homography, and searches for the solution that has the best agreement with all remaining matching pairs. This best agreement is found by calculating the mapping error between each transformed SIFT point of an input image and its corresponding SIFT point of the Master Image. To eliminate the bias towards any particular set of points, the best matching image-to-image alignment is achieved by recalculating the homography using all features with a small mapping error of the best homography found by RANSAC. Figure 9 shows an example of an image that is aligned to a corresponding Master Image.

**3.4  Automatic snow classification**

We perform snow classification experiments using the methods proposed by Salvatori et al. (2011) and Härer et al. (2016). The method by Salvatori et al. (2011) analyses the blue band digital number frequency histogram to set a snow threshold $DN_b$. First, the frequency histogram is smoothed using a moving average window of 5. The snow threshold is then automatically selected at the histogram's first local minimum above or equal to the intensity value 127. If no local minimum is found, the

snow threshold is set to the value 127. All pixel values equal or higher than this threshold value are classified as snow, whereas lower values are classified as snow-free.

    The second method is a snow classification routine included in PRACTISE V.2.1 (Härer et al., 2016). Since the method by Salvatori et al. (2011) only works reasonably well for non-shadowing areas (Härer et al., 2016; Arslan et al., 2017), this routine additionally detects snow in the shaded regions of an image. As a first step, the method of Härer et al. (2016) applies the blue-

band classification proposed by Salvatori et al. (2011). In a second step, Härer et al. (2016) refine snow classification using PCA for separating shaded snow cover from sunlit rock surfaces. Standardized RGB values in PCA space (PC score matrix) are calculated by multiplying the standardized RGB values (mean of 0 and standard deviation of 1) with the Principal Component (PC) coefficient matrix (calculated using singular value decomposition). The PC score matrix is normalized by scaling its values between 0 and 1. The first PC explains the largest variance in the data, but its frequency histogram is essentially identical to

the blue band frequency histogram. Therefore, Härer et al. (2016) use the frequency histograms of the second and third PC ($PC_2$ and $PC_3$) for separating shaded snow cover from other surfaces. The pixels are classified as snow if the following two conditions are fulfilled:

$$PC_3 < PC_2 \quad \text{and} \quad DN_b \geq DN_h \geq 63\,. \tag{9}$$

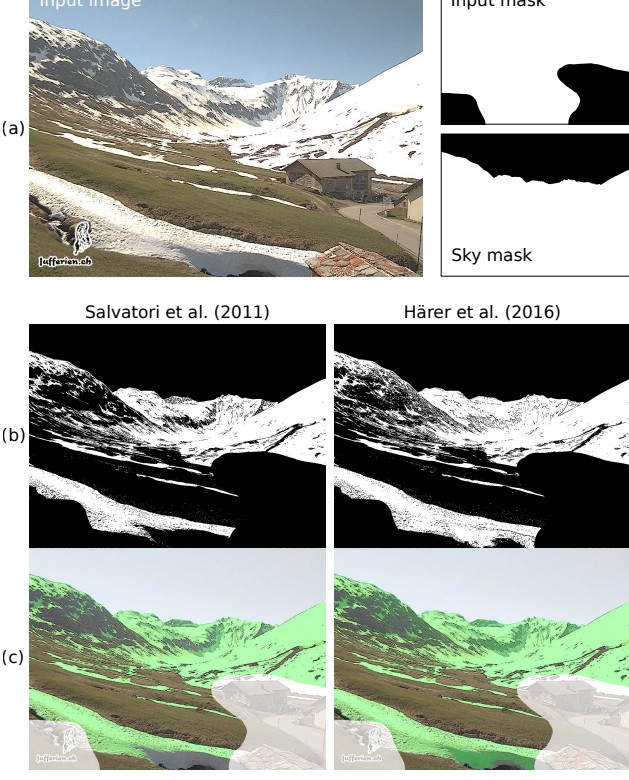

**Figure 10.** Example of a webcam image that is masked for subsequent snow classification using an input mask and a sky mask derived from the extracted mountain silhouette (a). Snow classification is applied using the methods by Salvatori et al. (2011) and Härer et al. (2016). Detected snow is shown (b) in white in the binary output image (black: no snow or masked out) and (c) as transparent green layer on the original webcam image (white transparent layer: masked region).

$DN_h$ is an additional condition to exclude very dark pixel values in the blue band channel (values $< 63$) since Härer et al. (2016) identified them as prone to snow misclassification. Moreover, blue-band values with a higher value than $DN_b$ are not considered either since they have been already identified as snow cover by the blue-band classification in the first step. As a third and fourth step, the method additionally identifies sunny rocks and calculates snow probability values for all the pixels that were not classified as snow in the first three steps.

We apply this snow classification by classifying all pixels detected in the first and second step as 'snow' and remaining pixel values as 'no snow'. A snow classification example is shown in Fig. 10. The snow classification takes as input a webcam image, the corresponding input mask described in Sect. 3.1, and a sky mask where all sky pixels are automatically masked out using the mountain silhouette extracted from the Master Image (see Fig. 10 (a)). The detected snow pixels by the method of Salvatori et al. (2011) and Härer et al. (2016) are shown in white (Fig. 10 (b)) and as green transparent layer (Fig. 10 (c)).

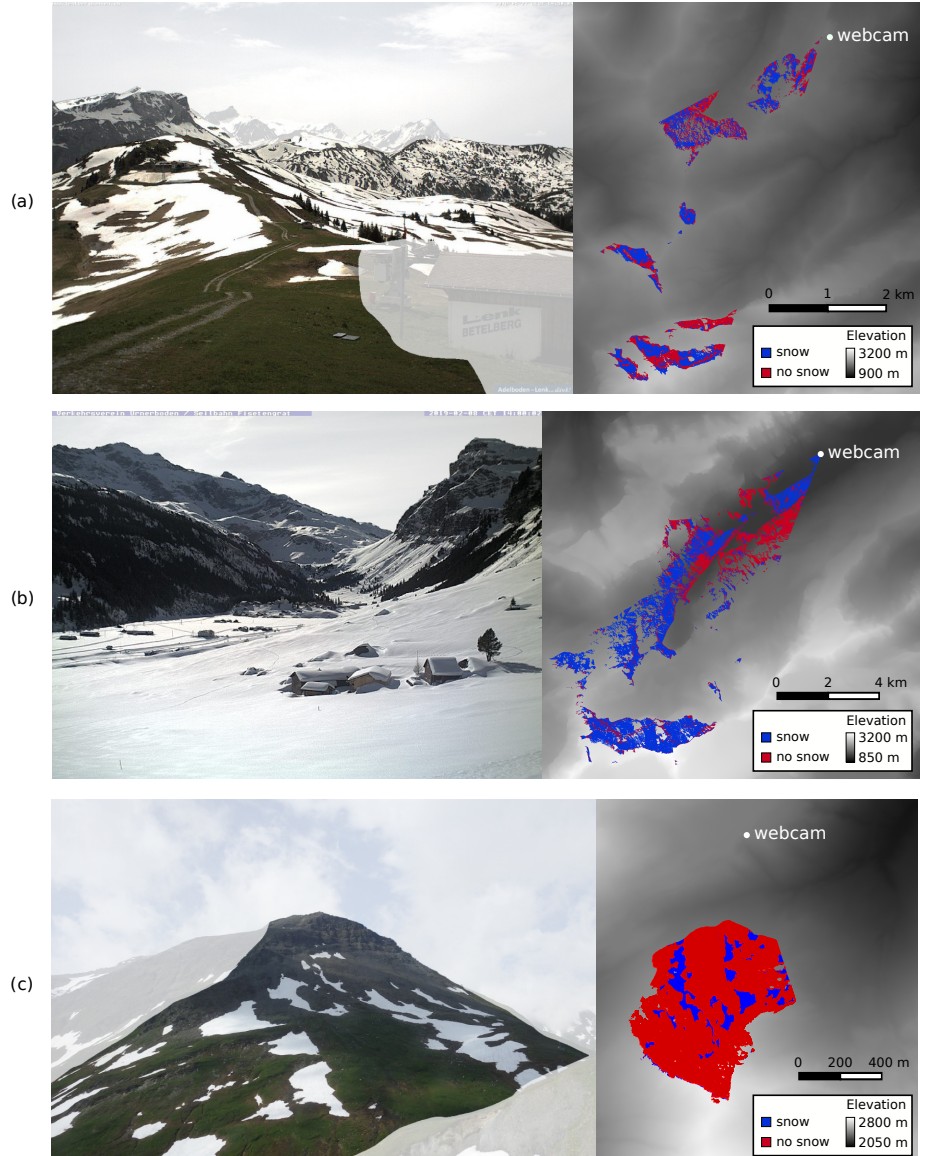

**Figure 11.** Example webcam images and resulting snow cover maps of three webcams in the (a) Lenk, (b) Urnerboden, and (c) Furkapass regions. Snow is classified using the method proposed by Salvatori et al. (2011). The white transparent layer on the webcam images shows the masked regions. The grayscale values of the snow cover maps show the elevation values of the area that is not visible from the webcam's location.

## 4 Snow cover maps

The transformation matrix found for each Master Image is used to generate a look-up table that relates all visible DEM grid cells to the associated image pixel. For each DEM grid cell in this look-up table, the associated classification result (i.e. 'snow'

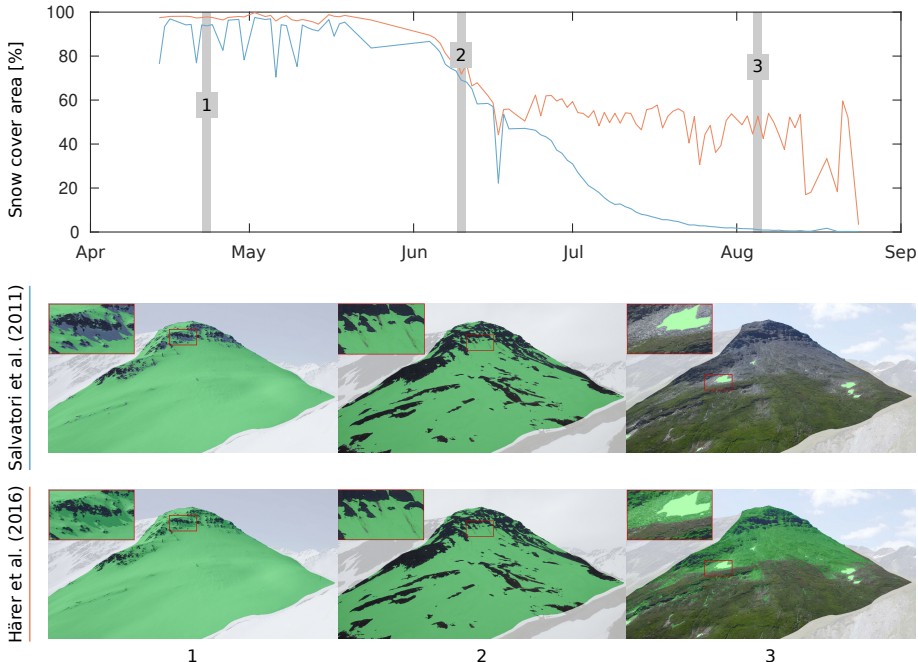

**Figure 12.** Percentage of snow covered area on a mountain hill in the Furkapass region from 14 April to 28 August 2015 using the snow classification proposed by Salvatori et al. (2011) (blue line) and Härer et al. (2016) (red line).

or 'no snow' of the classified webcam image pixels) is set, which results in a snow cover map of 2 m spatial resolution. Figure 11 shows three webcam images and resulting snow cover maps in the (a) Lenk, (b) Urnerboden, and (c) Furkapass regions.

Our procedure facilitates snow cover analyses using public webcams, as long as location of the camera can be estimated and a mountain silhouette is visible in the image. Figure 12 reveals the percentage of snow covered area on a mountain hill in the Furkapass region from 14 April to 28 August 2015 and three example images with applied classifications. Webcam images containing fog or adverse cloud cover that impede the view on the terrain were manually removed before processing. The differences caused by the two classification methods are discussed in Sect. 6.

## 5   Evaluation

In total, we apply image-to-DEM registration on 50 webcams. Our silhouette extraction technique successfully detects all 50 silhouettes. For five webcams, automatic image-to-DEM registration fails to find the appropriate orientation of the camera. This failure is either caused by heavy lens distortions of the camera system or due to several excerpts of similar looking mountain silhouettes that lead to a wrong orientation estimate on scale $k = 0$. In this section we evaluate the precision of the mapping between webcam image pixel coordinates and DEM coordinates, which we call mapping accuracy. This accuracy depends on

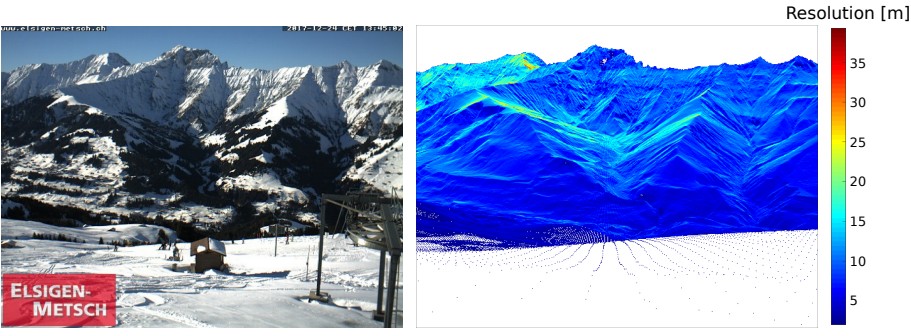

**Figure 13.** Example of projected pixel resolution for a webcam at Metschalp.

(1) uncertainties caused by the projection of low resolution webcam images on a high-resolution DEM (projection uncertainty), and (2) the ability of the registration approach to find the correct silhouette pair (registration accuracy).

## 5.1   Projection Uncertainty

Depending on the distance of the terrain to the webcam, the slope and aspect of the terrain, the webcam image resolution,
and its FOV, an image pixel is mapped onto one or several DEM grid cells. Therefore, image pixels are either upsampled or downsampled to the DEM's pixel resolution (2m). An approximation of the projected image pixel resolution can be calculated as root of the number of DEM grids an image pixel is mapped on times the resolution of the DEM (i.e. 2 m), assuming that an image pixel is mapped onto a rectangular region of DEM grids. Figure 13 shows the approximated projected pixel resolution of an example webcam image at Metschalp. The webcam image has an image resolution of $640 \times 480$ pixels and a horizontal
FOV of $47°$. In general, the projected pixel resolution close to the webcam is high and decreases with increasing distance to the webcam position. Moreover, the projected pixel resolution depends on the orientation of the slope with respect to the viewing direction. It is high for slopes orthogonal to the viewing direction and low at grazing angles near silhouettes. The mean projected pixel resolution found for 45 webcams is 4.5 m with a standard deviation of 4.4 m. If only DEM grids within a distance of 20 km to the webcam are considered, the mean projected pixel resolution increases to 2.9 m with a standard
deviation of 1.5 m.

## 5.2   Registration accuracy

To evaluate the accuracy of our automatic image-to-DEM registration, we select 20 webcams that comprise different areal extents and lens characteristics. Depending on the presence of structural image content, we manually select 5 to 15 GCPs per webcam using the SWISSIMAGE orthophoto. For 142 GCPs in total, we compute relative pixel errors (image space
distances) and the root mean square error (RMSE) of the distance between the real and projected GCPs in world coordinates (see Table 1). We differentiate between standard lens webcams (FOV $< 48°$) and wide-angle lens webcams (FOV $\geq 48°$). The relative pixel error is calculated as the distance between the pixel coordinate of a GCP and its pixel coordinate predicted by

**Table 1.** Projection error of ground control points (GCPs) in standard lens (FOV $< 48^\circ$) and wide-angle lens (FOV $\geq 48^\circ$) webcam images.

| | #cams | #GCPs | GCP RMSE [m] | Minimum residual [m] | Maximum residual [m] | $\sigma$ RMSE [m] | Relative pixel error [%] |
|---|---|---|---|---|---|---|---|
| All GCPs | 20 | 142 | 23.70 | 2.00 | 98.48 | 17.06 | 0.74 |
| GCPs standard lenses | 14 | 96 | 14.10 | 2.00 | 34.97 | 8.67 | 0.61 |
| GCPs wide-angle lenses | 6 | 46 | 36.31 | 2.03 | 98.48 | 23.63 | 1.00 |

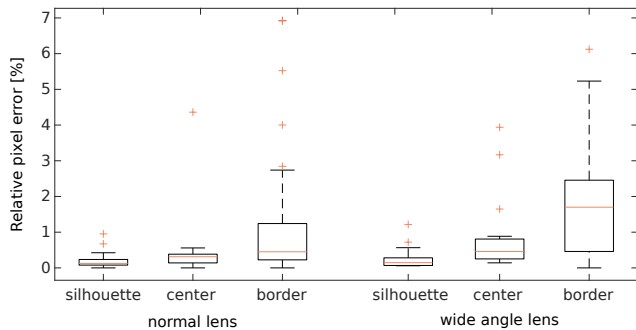

**Figure 14.** Relative pixel error of ground control points (GCPs) of standard and wide-angle lens webcams. Results are grouped in GCPs located at the mountain silhouette, the center region of the image, and the border region of the image (the outer 25 % of the total image width and height).

the transformation matrix. We report this distance as percentage of image diagonal. It is a measure to calculate the accuracy of our automatic image-to-DEM registration. Results show that the relative pixel error is higher for GCPs of wide-angle webcams than for GCPs of standard lens webcams (1% and 0.61%, respectively). This difference is mainly caused by lens distortions, which increase with a larger FOV and therewith lead to a discrepancy of the silhouette matching, mainly at the outer part of the

5 images. This discrepancy is even more prominent when considering the relative pixel error by comparing GCPs at the mountain silhouette, GCPs that are close to the image border (the outer 25% of the total image width/height), and the remaining GCPs in the center region of the image (see Fig. 14). GCPs at the silhouette indicate how well the image-to-DEM registration matches the two silhouettes. The further away GCPs are from the silhouette and the central part of the image, the more they are affected by the camera model used for image-to-DEM registration. Therefore, GCPs close to the image border are affected the most by

10 effects of lens distortions. The relative pixel error is notably higher for GCPs at the border of the images than the remaining GCPs, especially for wide-angle lens webcams. Not surprisingly, smallest errors are found for GCPs located at the mountain silhouette, since this silhouette is used for image-to-DEM registration.

The root mean square error (RMSE) of the distance between the real and projected GCPs in world coordinates is shown in Table 1. We find again a significant difference in the registration accuracy between webcams equipped with standard lenses

and wide-angle lenses. Registration accuracy reveals an overall RMSE of 23.7 m, with a RMSE of 14.1 m for standard lens

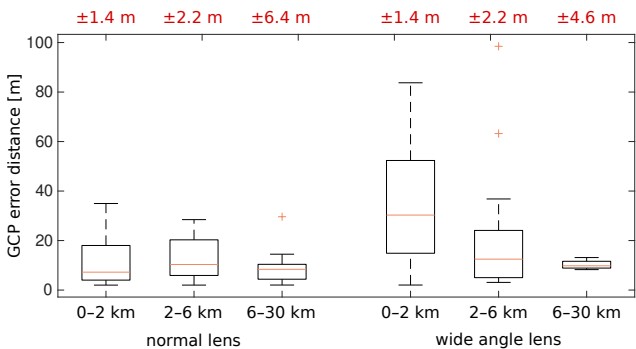

**Figure 15.** Distance error of the real and projected ground control points (GCPs) for standard and wide-angle lens webcams. Results are grouped in GCPs within 0–2 km, 2–6 km, and 6–30 km distance to the webcam. Median projection uncertainties are shown as red numbers on top of the figure.

webcams and 36.3 m for wide-angle lens webcams. We calculate the GCP error distance in world coordinates by projecting the registered pixels onto a map using the transformation matrix. In Fig. 15, box plots of the error distances between the real and projected GCPs are shown for standard and wide-angle lens webcams. Results are grouped into three categories of GCPs within 0–2 km, 2–6 km, and 6–30 km distance to the webcam. Since it is more difficult to set GCPs in low resolution webcam images, we use large structural features such as mountain peaks to set GCPs far away from the webcams. This ensures that we can select the appropriate pixel where the given GCP is actually located. We use the transformation matrix to project this pixel to world coordinates and, thus, we assume that this GCP is located in the center of the pixel. However, we have to take into account that this is not necessarily the real position of the GCP within the image pixel. As shown in Sect. 5.1, pixel values are mapped onto a certain area on a map. Therefore, we calculate the projection uncertainty of a GCP as ± the radius of the bounding volume of the DEM grids where the selected image pixel is projected on. We use the median to quantify the projection uncertainty of a group of GCPs. Median projection uncertainties are shown as red numbers on top of Fig. 15. It can be clearly seen that the largest error distances are caused by GCPs of wide-angle lens webcams that are located close to the webcam (0–2 km) and that the errors are generally lower further away from the webcams. For standard lens webcams, there is no considerable difference in the error distance between GPCs within 0–2 km, 2–6 km, and 6–30 km distance to the webcam. Even though projection uncertainties are higher for GCPs located further away from the webcams, for both, standard lens and wide-angle lens webcams, the mapping accuracy of GCPs that are more than 6 km away from the webcam (mean error distances of 8.6 m and 10.2 m with uncertainties of ±6.4 m and ±4.6 m, respectively) is comparable to the mapping accuracy found for GCPs within 0–6 km distance of normal lens webcams and GCPs within 2–6 km distance of wide-angle lens webcams.

## 6  Discussion

The performance of our automatic image-to-DEM registration procedure is promising. With marginal manual user input, we transform a webcam image into a georeferenced map. With an overall RMSE of about 23.7 m, our method is precise enough to validate or complement satellite-derived snow cover maps and offers snow cover analyses with a high spatio-temporal resolution over a large area. However, projection uncertainties have to be taken into account as well since they may highly differ depending on the selected webcam. We expect a lower performance of our image-to-DEM registration compared to approaches where camera parameters are available or GCPs are used to align an image to a DEM. However, having access to intrinsic and extrinsic camera parameters, or measuring these parameters using GCPs is infeasible for a reasonably large-scale camera network. The large differences of RMSE between standard lens webcams and wide-angle lens webcams suggest a further improvement of our camera model to account for lens distortions. Given the large amount of webcams, we can also exclude webcams equipped with wide-angle lenses from analyses to notably reduce mapping errors (RMSE of 14.1 m found for 14 webcams equipped with standard lenses, see Table 1). Another solution is to use only the central part of an image if the FOV of the webcam is higher than a certain threshold.

Our method relies on a precise estimation of the webcam location. Especially when a decreasing slope is visible in the near field of the webcam, significant mapping errors may occur. For example, a lower estimate of the installation height may cause a pixel in 10 m distance to be mapped onto the counter slope 2 km away. Therefore, we recommend to mask out regions that are on the same slope as the webcam itself or areas close to edges with huge depth differences. Since we did not measure the ground truth location of the selected webcams, a direct evaluation of the estimated location accuracy is not possible. However, we roughly estimate an accuracy of about 5 m by leveraging the orthophoto SWISSIMAGE and prior knowledge about the approximate webcam location (for instance, mounted on a specific wall of a building).

In general, we propose to mask out regions that are close to the webcam to avoid large mapping errors as shown in Figure 15 for webcams with wide-angle lenses. These large mapping errors may be caused by an imprecise location estimation. However, this effect was not observed for standard lens webcams. Hence, the large mapping errors close to the webcam can be attributed to the fact that close GCPs are generally more often located at the outer part of the image where lens distortions increase. In addition, areas closer to a webcam may generally have a larger mapping error as only the mountain silhouette is used for the image-to-DEM registration. Therefore, these areas are additionally affected by the selected camera model used for image-to-DEM registration. Additionally, we propose to exclude regions that are far away from the webcam (i.e. > 15 or 20 km) to avoid large projection uncertainties and to ensure a high spatial resolution. Moreover, it has to be taken into account that projection uncertainties may strongly increase if the slope and aspect of the DEM grid with respect to the viewing direction is high. For single image pixels, projection uncertainty can be extremely high if the pixel is mapped onto several non-adjacent DEM grids (e.g. if a pixel is projected onto DEM grids on a hill or peak as well as on the DEM grids behind the hill on the counter slope).

For most webcams, an intentional, significant change in its orientation occurs only occasionally and therefore, a landscape can be analyzed over a long time period in the case of an available image archive. Our image-to-image alignment enables to precisely correct small changes in orientation of webcam images and works generally well for images with similar image

content. Alignment artifacts from e.g. logos in the image are eliminated by using RANSAC. Since some errors may occur if the image content differs too much, we propose not to align snowy winter images to snow-free images scenes and vice versa.

The snow classification method proposed by Salvatori et al. (2011) is frequently used and discussed in recent studies. Many of these studies emphasize the problem of misclassification due to snow in shadowing regions (e.g. Härer et al., 2016; Arslan et al., 2017; Salzano et al., 2019). We have observed the same issue, especially for winter scenes with a low solar zenith angle. The comparison with the snow classification method proposed by Härer et al. (2016) reveals a similar pattern for all the processed webcams: The method by Salvatori et al. (2011) is underestimating snow cover, mainly in shadowing areas (see Fig. 12 for an example). For snowy winter scenes, the PCA method by Härer et al. (2016) performs very well and is able to correctly classify snow cover in shadowing areas. However, once less than about 50% of snow is present in an image, the method overestimates snow cover and classifies rock, trees or grass as snow (see Fig. 12). This is often observed when no shaded snow cover is present or in the case of strong illumination conditions. As shadows from structural terrain become less in spring, the method of Salvatori et al. (2011) often only weakly underestimates the snow cover. For rare cases of very low illumination conditions, both methods fail to correctly classify snow. In our framework, we use a combination of both methods to get the best possible snow classification result. However, there is a need for an improved snow classification method. This method should be able to classify snow under varying illumination conditions and ideally can distinguish between snow and clouds or fog.

The differentiation between snow, clouds, and fog currently remains an unsolved problem for RGB images. Even though webcams are often located below the cloud cover, low clouds and fog in front of the landscape are manually removed to not falsify snow classification. Images containing fog and clouds on a substantial part of the image could be automatically removed by comparing the edges of a cloud free image with edges of a potentially cloud covered image. However, clouds and fog that impede the view on a smaller part of the landscape are difficult to distinguish from snow. A possible method to remove such cloud cover is, for example, to aggregate all the images collected by a webcam in a day as proposed by Fedorov et al. (2016). However, the aggregated images may loose contrast and contain mixed pixel information, which in turn will affect snow classification. Moreover, long-lasting cloudy conditions may remain undetected by this approach and the aggregation will lower the temporal resolution. Therefore, we consider to investigate cloud and fog detection in webcam images for future work.

Since our image-to-DEM registration requires a visible mountain silhouette, it is not suited for webcams that observe flat areas. Moreover, there are geographical limitations since webcams might not be installed in very remote areas. Generally, a large-scale coverage of a region might be only possible in countries with a well-developed infrastructure. Nevertheless, the high number of freely available webcams worldwide combined with our semi-automatic procedure offers a unique potential to complement and evaluate satellite-derived snow cover information. For example, our webcam snow cover maps may facilitate the gap-filling of partly cloud-obscured satellite-based snow cover maps or improve snow classification in steep terrain or shadow-affected image scenes.

## 7 Conclusions

We present a semi-automatic procedure to derive snow cover maps from freely available webcam images in the Swiss Alps. Our registration approach automatically estimates webcams' parameters, which allows to relate pixels of a webcam image to their real-world coordinates. Additionally, we use a method for automatic image-to-image alignment and compare two recent snow classification methods. A detailed evaluation of the automatic georectification is carried out and reveals in a RMSE of 23.7 m, with a RMSE of 14.1 m for webcams equipped with standard lenses and 36.3 m for webcams equipped with wide-angle lenses. To the best of our knowledge, no other method is able to offer this accuracy on such a high spatio-temporal resolution over a large area. Large accuracy differences between standard lens webcams and webcams equipped with wide-angle lenses suggest to improve our camera model to incorporate effects of lens distortions or to use only the central part of an image to generate more accurate snow cover maps. However, an improvement of RGB snow classification is essential to automatically derive snow cover maps, i.e. to avoid the manual removal of cloudy scenes. Nevertheless, our approach offers snow cover analyses with a high spatio-temporal resolution over a large area with a minimum of manual user input. Our webcam-based snow cover monitoring network could not only serve as a reference for improved validation of satellite-based approaches, but also complement satellite-based snow cover retrieval. As an example, webcam-based snow cover information could be used to improve gap-filling methods to eliminate cloud cover in satellite-based snow cover products. Especially in spring during the snowmelt period, webcams could help to detect snow that may fall and melt within several days during cloudy conditions. In addition, our webcam-based snow cover product can be used to validate Sentinel-2 and Landsat based snow cover products. We are therefore planning to extend our webcam archive with additional webcams located in the European Alps. Finally, our procedure, in particular the snow/cloud classification, could be improved to enable semi-operational processing for a near real-time service, which could support federal agencies (e.g. MeteoSwiss, WSL-SLF) for their weather forecast activities or avalanche warning.

*Author contributions.* Céline Portenier developed the code and performed the data analysis with advice from Fabia Hüsler and Stefan Wunderle. Stefan Härer provided matlab code of the software PRACTISE. Céline Portenier wrote the manuscript with contributions from all co-authors.

*Competing interests.* The authors declare that they have no conflict of interest.

*Acknowledgements.* The digital elevation model swissALTI$^{3D}$ and the orthophoto SWISSIMAGE were obtained from the Federal Office of Topography (swisstopo). The authors acknowledge Kai Kobler for providing updated webcam images on www.kaikowetter.ch and all webcam owners that provide their images online, in particular Armin Rist and Sara Fischer. Further we gratefully acknowledge Simon Gascoin and Tiziano Portenier for their constructive comments on the manuscript.

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
