# Peer review of "Towards a webcam-based snow cover monitoring network: methodology and evaluation"

_The Cryosphere, 2019_

## Referee Comment (RC1) · Yves Bühler (Referee) · 29 Jul 2019

The paper entitled "Towards a webcam-based snow cover monitoring network: methodology and evaluation" by C. Portenier et al. presents an innovative and promising approach to exploit available webcam imagery for snow cover (SC) mapping in Switzerland. This is a first important step towards the combination of different sensors and platforms to monitor snow parameters over large regions with high temporal and spatial resolution. However, there are three main points I would like to see clarified and complemented before I can recommend the paper for publication:

1. Snow cover classification

[Figure]

Two quite simple methods are applied to classify snow covered areas in the webcam imagery (Salvatori et al. 2011 & Härer et al. 2016). This part is not complete in my opinion. The method by Härer et al. 2016 should be described in more detail, now there is just a reference to this paper. As the authors state themselves in the discussion and conclusion, there is a big potential for improvement concerning this point. As the snow cover classification is an essential part of the entire processing chain, I recommend to invested some more time to look at different other options. Federov et al. 2016 and Rüfenacht et al. 2014 already tested more advanced classification methods. The authors have at least to test and discuss these options and justify why they select the other options. I also suggest to overwork Fig. 10 including the results from all classification algorithms so they get comparable visible in an example image. Now only one method is demonstrated and it is not clear which one.

2. Geolocation accuracy assessment

In my understanding the spatial resolution of the imagery and with it the achievable accuracy is very much dependent on the distance of the camera to the terrain. The spatial resolution in your imagery must vary a lot! You do not really address this point. In contrast, your results even suggest that the accuracies get better with distance (as this is the area close to the maintain ridges that you used to co-register the image to the DEM, Fig. 13). Here clarification is needed. I would be interested to read what the image resolutions ranges are for the different webcams and what problems the varying resolutions cause. How does the resolution problem relate to the accuracy values you calculate?

3. Conclusions

The conclusions are too brief in my opinion. Here I would like to read a bit more of an outlook. How does it go on? For what satellite products will it be applied? Is there the intention to go also to other countries with this method? Please extend the conclusions.

Detailed comments:

P2L17: I would be careful to talk about very high spatial resolution monitoring. Only the regions close by the camera are highs spatial resolution (0.1 – 2 m). Further away it gets much coarser. Maybe you can define what you understand by very high spatio-temporal scales.

P2L19: You could be a bit more precise here, when there is fog there will be no information. What types of clouds will still be OK as also the contrast will be lowered by high clouds. I see the big benefit of the method for the evaluation of satellite products not only for complementation maybe you can add that.

P3Fig1: Please be careful about the publication of swisstopo data. Do you have all necessary rights? If so you should have a specific contract number from swisstopo which allows you to publish it.

P3L13: Only 57% of all cameras fulfill your conditions. Could you please explain a bit more here why? Are there options to increase that ration?

P4L6: The current resolution of swissimage is now 10 cm in the lowlands and 25 cm in the Alps

P5L7: How do you estimate the accuracy of the location estimation?

P9L22: why do cameras change their orientation? How often does that happen? Please explain

P11L32: How are "bad images" detected and eliminated? Is it done manually? If yes, would there be options to automatically detect "bad images". This is an important point as there will be many images that should be removed in long timelines. I would like to see some more details on this point.

P12Fig9: Here you choose a fully snow-covered scene as example. In my opinion it would be of much more interest for the readers to see this demonstration on a partially snow covered scene. Could you change that?

P14L10: Here you state that the best accuracy is close to the mountain ridge. But these are the regions with low spatial resolution. How accurate are the other points (see my main point N°2).

P14L14: here you state "residuals generally decrease with the distance to the webcam". From my understanding they should increase in that direction as it is much more difficult to find and set GCP's far away on lower resolution imagery. Please clarify.

P16L24: Please explain a bit more what RANSAC is and how you apply it.

References:

Fedorov, R., Camerada, A., Fraternali, P., and Tagliasacchi, M.: Estimating Snow Cover From Publicly Available Images, IEEE Transactions on Multimedia, 18, 1187-1200, 10.1109/tmm.2016.2535356, 2016.

Härer, S., Bernhardt, M., and Schulz, K.: PRACTISE – Photo Rectification And ClassificaTIon SoftwarE (V.2.1), Geoscientific Model Development, 9, 307-321, 10.5194/gmd-9-307-2016, 2016.

Rüfenacht, D., Brown, M., Beutel, J., and Süsstrunk, S.: Temporally consistent snow cover estimation from noisy, irregularly sampled measurements, 2014 International Conference on Computer Vision Theory and Applications (VISAPP), 2014, 275-283,

---

## Referee Comment (RC2) · A.N. Arslan (Referee) · 4 Aug 2019

**Review comments** on tc-2019-142 manuscript, entitled," Towards a webcam-based snow cover monitoring network: methodology and evaluation".

**General comments:**

The tc-2019-142 manuscript, entitled," Towards a webcam-based snow cover monitoring network: methodology and evaluation" presents a semi-automatic approach procedure to derive snow cover maps. The semi-automatic approach procedure is consist of (1) automatic image to image alignment and (2) automatic image to DEM registration which are the contributions of the manuscript. In addition a snow classification method (two existing methods in literature presented) and a manual user input (for estimation webcam's location) are needed for estimating snow cover from a webcam image.

The purpose of the work is clearly articulated and the methodology and results are adequately presented.

**Specific Comments:**

There are following issues which I believe need more discussions such as

(1) webcam-based snow cover monitoring network

(2) Arbitrary images to generate snow cover maps

(3) Most of existing studies use single cameras and thus are limited in areal coverage. In particular, they either require known camera parameters (i.e., extrinsic and intrinsic camera parameters such as the camera orientation or the FOV of the camera) or require significant manual user input (e.g., ground control points (GCPs)) to georectify terrestrial photography

(4) Since camera parameters are not readily available for public webcams, and manually setting GCPs for a large number of cameras is time-consuming, these methods are of limited application for our purposes.

(1) webcam-based snow cover monitoring: This is very good concept. It is very good to explain this concept in more detail and how the proposed methodology can be applied and what current status of existing webcam networks is. What should be done apart from improving snow classification methods mentioned in the manuscript?

(2) I am not sure about arbitrary images as one should know the location of camera. May be this is a bit misleading?

(3) It is not clear for me what differences are! In the proposed procedure in the manuscript one has to create "master image" How more easy and accurate is creating master image than procedures in the existing studies?

(4) How about camera locations? How do one get locations of webcams which are need as input in the proposed manuscript? As the objective is "towards webcam-based snow cover monitoring" why not setting GCPs for time-consuming.  The creating an accurate master image is an essential part of the proposed work in the manuscript.

How time consuming is creating a good master image? What is applicability of creating master image in various environment as silhouette extraction is based on the assumption that the mountain silhouette in the manuscript. How about open and forested areas isn't it big limitations of the method towards webcam-based snow cover monitoring network? That's why all should be explained!

There is an evaluation on the accuracy of the automatic image-to-DEM registration.  There is no an evaluation of the proposed procedure, entitled, "a semi-automatic approach procedure".

As it is mentioned at the end of the discussion in the manuscript "our webcam snow cover maps facilitate the gap filling of partly cloud-obscured satellite-based snow cover maps or improve snow classification in steep terrain 15 or shadow-affected image scenes." It would be good to see some evaluation of the proposed procedure supporting this statement.

**Technical Corrections:**

In Figure 11: It is good to explain colors like red and blue; which one is Salvotori et.al method and etc.

---

## Author Comment (AC1) · 28 Sep 2019

We would like to thank Yves Bühler for this careful and detailed review that was helpful to improve the manuscript.

Below we respond to all comments by Yves Bühler. The responses (normal font style) are following the *referees' comments* (displayed in italic font style) directly.

*The paper entitled "Towards a webcam-based snow cover monitoring network: methodology and evaluation" by C. Portenier et al. presents an innovative and promising approach to exploit available webcam imagery for snow cover (SC) mapping in Switzer-*

[Figure]

*land. This is a first important step towards the combination of different sensors and platforms to monitor snow parameters over large regions with high temporal and spatial resolution. However, there are three main points I would like to see clarified and complemented before I can recommend the paper for publication:*

*1. Snow cover classification*
*Two quite simple methods are applied to classify snow covered areas in the webcam imagery (Salvatori et al. 2011 Härer et al. 2016). This part is not complete in my opinion. The method by Härer et al. 2016 should be described in more detail, now there is just a reference to this paper. As the authors state themselves in the discussion and conclusion, there is a big potential for improvement concerning this point. As the snow cover classification is an essential part of the entire processing chain, I recommend to invested some more time to look at different other options. Federov et al. 2016 and Rüfenacht et al. 2014 already tested more advanced classification methods. The authors have at least to test and discuss these options and justify why they select the other options. I also suggest to overwork Fig. 10 including the results from all classification algorithms so they get comparable visible in an example image. Now only one method is demonstrated and it is not clear which one.*

We present an overall framework for the processing of webcam images with a snow classification module. We agree that snow classification is an essential part of our processing chain. We will include a more detailed description of the method by Härer et al. 2016 and discuss other methods used for RGB snow classification to give the reader a broader overview on existing snow classification approaches.

As stated in the work by Rüfenacht et al. (2014), their proposed method is not able to detect snow in shadowed areas of sunny scenes. The authors avoid this issue by explicitly excluding respective images from their analysis. We did therefore not consider applying their approach to our problem, since such situations often occur in our application. Hence, we leveraged the method proposed by Härer et al. (2016), since it is explicitly designed to work under such difficult conditions and is therefore a

better fit for our application.

We agree that the method proposed by Fedorov et al. (2016) is a promising solution for our application and we will try to include a comparison to their method. Since they trained a machine learning model to obtain their snow classification framework, we find that it is not feasible to reproduce their model without having access to the respective training data. We contacted the authors in order to apply their learned model on our data, or to at least retrain their model with their respective training data.

It is correct that we emphasize the need for an improved classification, since we found that none of the existing methods work reasonably robust for our application, in particular the distinction between snow and clouds as well as snow classification under difficult illumination conditions. We consider to investigate this direction as future work.

*2. Geolocation accuracy assessment*
*In my understanding the spatial resolution of the imagery and with it the achievable accuracy is very much dependent on the distance of the camera to the terrain. The spatial resolution in your imagery must vary a lot! You do not really address this point. In contrast, your results even suggest that the accuracies get better with distance (as this is the area close to the maintain ridges that you used to co-register the image to the DEM, Fig. 13). Here clarification is needed. I would be interested to read what the image resolutions ranges are for the different webcams and what problems the varying resolutions cause. How does the resolution problem relate to the accuracy values you calculate?*

Thank you for pointing this out. We agree that the spatial resolution of a webcam image has an impact on the resulting snow cover map. Depending on the distance of the terrain to the webcam, image pixels of the webcam are either upsampled or downsampled to the DEM's pixel resolution (2m). We will discuss this relationship in the text and provide an example for a typical webcam image where we estimate the resolution range of the projected image. In addition, we provide a histogram that shows

the distribution of webcam image resolutions in our archive.

We will provide resulting uncertainties for Fig.10 that occur due to the above mentioned effect. However, the uncertainties do not change the overall picture (for instance, the uncertainty for the GCPs within a distance of 6-30km is ±5.6m).

*3. Conclusions*
*The conclusions are too brief in my opinion. Here I would like to read a bit more of an outlook. How does it go on? For what satellite products will it be applied? Is there the intention to go also to other countries with this method? Please extend the conclusions.*

We agree that the conclusions are too brief. We will extend the conclusions with an outlook that provides further information on the potential application of our procedure. First, ESA CCI snow is using Sentinel-2 and Landsat data as validation source for MODIS and AVHRR snow cover fraction retrieval. To in turn evaluate the accuracy of Sentinel-2 and Landsat based products, our webcam-based snow monitoring product can be applied. Second, our product can further be used to validate Sentinel-2 and Sentinel-1 based snow cover products generated by Copernicus and Theia. In fact, Theia is highly interested to use our product for an accuracy study of Sentinel-2 snow cover maps. Third, our procedure, in particular the snow classification, could be improved to enable semi-operational processing for a NRT-service, which could support federal agencies (e.g. MeteoSwiss, WSL-SLF) for their weather forecast activities or avalanche warning.

*Detailed comments:*

*P2L17: I would be careful to talk about very high spatial resolution monitoring. Only the regions close by the camera are highs spatial resolution (0.1 – 2 m). Further away it gets much coarser. Maybe you can define what you understand by very high spatio-temporal scales.*

Thank you for pointing this out, we will clarify this in the text. As an example regarding
the spatial resolution, a pixel of a comparably low resolution webcam image ($640 \times 480$ pixels) imaging an area at 30km distance to the camera has a projected pixel size of less than 20m. This is comparable to e.g., Sentinel-2, thus we consider this high resolution. Moreover, since we know the distance to the camera on a per-pixel basis, we can even exclude areas that are too far away, which limits the worst case spatial resolution as desired. Regarding temporal resolution, most webcams record at least one image per hour, which we consider high temporal resolution. We will consistently replace the term 'very high' with 'high' in the text.

*P2L19: You could be a bit more precise here, when there is fog there will be no information. What types of clouds will still be OK as also the contrast will be lowered by high clouds. I see the big benefit of the method for the evaluation of satellite products not only for complementation maybe you can add that.*

We agree that our statement here is indistinct and we try to clarify hereby. As long as cloud cover and fog are above the mountain silhouette and therewith do not disturb the view on the ground, webcam images can potentially provide snow cover information. However, depending on the snow classification technique, reduced contrast due to overcast weather degrades classification accuracy. We think that more robust snow classification techniques can still be able to reliably classify snow under such conditions, and we consider investigating this in future work.

*P3Fig1: Please be careful about the publication of swisstopo data. Do you have all necessary rights? If so you should have a specific contract number from swisstopo which allows you to publish it.*

Thank you for pointing this out. We have the necessary rights and will recheck the correct indication of source.

*P3L13: Only 57% of all cameras fulfill your conditions. Could you please explain a bit more here why? Are there options to increase that ratio?*

We agree that this is not clearly stated in the text. Up to now, we estimated the locations of 297 webcams and the number constantly increases. It is not that only those cameras fulfill our conditions, and we will explain this point clearer. For instance, for the webcams provided by kaikowetter.ch, about 70% of the cameras satisfy our conditions. The other 30% either do not feature mountain silhouettes or the silhouettes are partially occluded by trees or buildings. Due to the nature of our method, such cameras cannot be used. We will provide this information in the revised version of the manuscript.

*P4L6: The current resolution of swissimage is now 10 cm in the lowlands and 25 cm in the Alps*

Thank you for pointing this out, we will correct this within the revision.

*P5L7: How do you estimate the accuracy of the location estimation?*

We did not measure the ground truth location for our webcams, therefore a direct evaluation of the estimated location is not possible. However, by leveraging the orthophoto SWISSIMAGE and prior knowledge about the approximate webcam location (for instance, mounted on a specific wall of a building), we could roughly estimate the accuracy. We will include these details in the revised text.

*P9L22: why do cameras change their orientation? How often does that happen? Please explain*

Most webcams are exposed to wind and therefore occasional tiny camera movements occur. Moreover, for few webcams major movements can occur due to human interaction, intentionally or unintentionally. While small movements due to the first reason occur frequently, the second case is rare, at most monthly. We will add this to the text.

*P11L32: How are "bad images" detected and eliminated? Is it done manually? If yes, would there be options to automatically detect "bad images". This is an important point as there will be many images that should be removed in long timelines. I would like to see some more details on this point.*

We remove "bad images" manually. Automatic snow/cloud distinction is still an unsolved problem, hence automatic detection of unusable images is difficult. Other automatic approaches based on temporal smoothing limit the temporal resolution, which we want to avoid. However, we consider investigating automatic techniques for future work. We will discuss this in the revised text.

*P12Fig9: Here you choose a fully snow-covered scene as example. In my opinion it would be of much more interest for the readers to see this demonstration on a partially snow covered scene. Could you change that?*

We will replace the example in Fig. 9.

*P14L10: Here you state that the best accuracy is close to the mountain ridge. But these are the regions with low spatial resolution. How accurate are the other points (see my main point N°2).*

As mentioned above (answer to main point N°2), the uncertainty indeed increases with increasing distance to the camera. However, as mentioned this uncertainty is still much lower than the difference in the estimated residuals, therefore the overall statement still holds.

*P14L14: here you state "residuals generally decrease with the distance to the webcam". From my understanding they should increase in that direction as it is much more difficult to find and set GCP's far away on lower resolution imagery. Please clarify.*

By leveraging the DEM, GCPs on the mountain ridge can actually be set quite accurately (modulo the distance dependent uncertainty discussed above). As promised in the answer to main point N°2, we will explain this much clearer in the text.

*P16L24: Please explain a bit more what RANSAC is and how you apply it.*

A detailed description of the RANSAC algorithm is provided within the methods section (Sect. 3.3, page 11).

References:

Fedorov, R., Camerada, A., Fraternali, P., and Tagliasacchi, M.: Estimating Snow Cover From Publicly Available Images, IEEE Transactions on Multimedia, 18, 1187-1200, 10.1109/tmm.2016.2535356, 2016.

Härer, S., Bernhardt, M., and Schulz, K.: PRACTISE ndash; Photo Rectification And ClassificaTIon SoftwarE (V.2.1), Geoscientific Model Development, 9, 307-321, 10.5194/gmd-9-307-2016, 2016.

Rüfenacht, D., Brown, M., Beutel, J., and Süsstrunk, S.: Temporally consistent snow cover estimation from noisy, irregularly sampled measurements, 2014 International Conference on Computer Vision Theory and Applications (VISAPP), 2014, 275-283.

---

## Author Comment (AC2) · 28 Sep 2019

We thank A. N. Arslan for his valuable and constructive comments that were helpful to improve the manuscript.

Below we respond to all comments by A. N. Arslan. The responses (normal font style) are following the *referees' comments* (displayed in italic font style) directly.

*General comments:*
*The tc-2019-142 manuscript, entitled," Towards a webcam-based snow cover monitoring network: methodology and evaluation" presents a semi-automatic approach proce-*

*dure to derive snow cover maps. The semi-automatic approach procedure is consist of (1) automatic image to image alignment and (2) automatic image to DEM registration which are the contributions of the manuscript. In addition a snow classification method (two existing methods in literature presented) and a manual user input (for estimation webcam's location) are needed for estimating snow cover from a webcam image.*

*The purpose of the work is clearly articulated and the methodology and results are adequately presented.*

*Specific Comments:*
*There are following issues which I believe need more discussions such as*
*(1) webcam-based snow cover monitoring network*
*(2) Arbitrary images to generate snow cover maps*
*(3) Most of existing studies use single cameras and thus are limited in areal coverage. In particular, they either require known camera parameters (i.e., extrinsic and intrinsic camera parameters such as the camera orientation or the FOV of the camera) or require significant manual user input (e.g., ground control points (GCPs)) to georectify terrestrial photography*
*(4) Since camera parameters are not readily available for public webcams, and manually setting GCPs for a large number of cameras is time-consuming, these methods are of limited application for our purposes.*

*(1) webcam-based snow cover monitoring: This is very good concept. It is very good to explain this concept in more detail and how the proposed methodology can be applied and what current status of existing webcam networks is. What should be done apart from improving snow classification methods mentioned in the manuscript?*

Thank you for pointing this out. We agree to add more details about the possible applications of using webcam images for snow cover monitoring. We will extend the discussion on possible applications and improvements of our methodology.

*(2) I am not sure about arbitrary images as one should know the location of camera.*

*May be this is a bit misleading?*

We agree that this is a bit overstated. We will remove the word 'arbitrary' in the revision of the manuscript, and mention that the location must be estimateable.

*(3) It is not clear for me what differences are! In the proposed procedure in the manuscript one has to create "master image" How more easy and accurate is creating master image than procedures in the existing studies?*

Selecting a single Master image per webcam is straightforward, the only assumption is that the daytime and weather conditions are such that the mountain silhouette is clearly visible. In contrast, having access to intrinsic and extrinsic camera parameters, or measuring these parameters using GCPs is infeasible for a reasonably large-scale camera network. Since our method computes these parameters using only the Master image and camera position estimation as input, it is feasible to compile a large-scale camera network with our approach. However, it is true that the accuracy of our image-to-DEM mapping is expected to be lower compared to approaches where ground truth camera parameters are available. We will discuss this in the revised text.

*(4) How about camera locations? How do one get locations of webcams which are need as input in the proposed manuscript? As the objective is "towards webcam-based snow cover monitoring" why not setting GCPs for time-consuming. The creating an accurate master image is an essential part of the proposed work in the manuscript.How time consuming is creating a good master image? What is applicability of creating master image in various environment as silhouette extraction is based on the assumption that the mountain silhouette in the manuscript. How about open and forested areas isn't it big limitations of the method towards webcam-based snow cover monitoring network? That's why all should be explained!*

We manually estimate webcam locations by considering the position of objects visible in the webcam image, the orthophoto SWISSIMAGE, and additional information provided by the webcam owner. This can be, for example, the name of a restaurant

(Section 3.1). It is correct that webcams cannot be used by our approach if they do not feature mountain silhouettes due to open or forested areas, or where the silhouette is partially occluded by trees or buildings. We will discuss this in the revised version of the manuscript.

*There is an evaluation on the accuracy of the automatic image-to-DEM registration. There is no an evaluation of the proposed procedure, entitled, "a semi-automatic approach procedure".*

We agree that the main focus of our evaluation lies on the automatic image-to-DEM registration, which we consider our main contribution. We did not explicitly evaluate parts of our pipeline that we adopted. However, we provide a qualitative comparison of the leveraged snow classification techniques.

*As it is mentioned at the end of the discussion in the manuscript "our webcam snow cover maps facilitate the gap filling of partly cloud-obscured satellite-based snow cover maps or improve snow classification in steep terrain or shadow-affected image scenes." It would be good to see some evaluation of the proposed procedure supporting this statement.*

We agree that it is an important further step to apply our proposed framework to perform such evaluations. However, this would be out of scope for the current work and we consider this as future work. We will mention this in the discussion section.

*Technical Corrections:*
*In Figure 11: It is good to explain colors like red and blue; which one is Salvotori et.al method and etc.*

Thank you for pointing this out. We will describe the color coding in the figure caption.

---

## Referee Report (RR1)

**Review comments** on tc-2019-142 manuscript, entitled," Towards a webcam-based snow cover monitoring network: methodology and evaluation".

**General comments**:

The tc-2019-142 manuscript, entitled," Towards a webcam-based snow cover monitoring network: methodology and evaluation" presents a semi-automatic approach procedure to derive snow cover maps. The semi-automatic approach procedure is consist of (1) automatic image to image alignment and (2) automatic image to DEM registration which are the contributions of the manuscript. In addition a snow classification method (two existing methods in literature presented) and a manual user input (for estimation webcam's location) are needed for estimating snow cover from a webcam image.

The purpose of the work is clearly articulated and the methodology and results are adequately presented.

**Minor Comments:**

**Comment 1**: I would recommend adding following references optional which I believe useful to audiences:

**Page 1**:  for in-situ measurements:

Pirazzini, R.; Leppänen, L.; Picard, G.; Lopez-Moreno, J.I.; Marty, C.; Macelloni, G.; Kontu, A.; Von Lerber, A.; Tanis, C.M.; Schneebeli, M.; De Rosnay, P.; Arslan, A.N. European In-Situ Snow Measurements: Practices and Purposes. Sensors 2018, 18, 2016.

Page 21: "In addition, our webcam-based snow cover product can be used to validate Sentinel-2 and Landsat based snow cover products"

It would be good to list earlier studies on webcam-based snow cover product which were used for validation satellite-derived products like Sentinel-2 and Landsat. There is one given below

Piazzi, G.; Tanis, C.; Kuter, S.; Simsek, B.; Puca, S.; Toniazzo, A.; Takala, M.; Akyürek, Z.; Gabellani, S.; Arslan, A. Cross-Country Assessment of H-SAF Snow Products by Sentinel-2 Imagery Validated against In-Situ Observations and Webcam Photography. Geosciences 2019, 9, 129.

**Comment 2:**

**Page 3, Line 13**: "They propose to apply the blue band classification by Salvatorietal. (2011) and subsequently use principal component analysis (PCA) to separate shaded snow cover from sunlit rock surfaces.."

This sentence is not linked to previous one and I believe should be rewritten like what is "they".. NOT CLEAR!

---

## Author Response (AR2)

**Response to review comments on tc-2019-142 manuscript, entitled," Towards a webcam-based snow cover monitoring network: methodology and evaluation".**

We thank Ketil Isaksen and A. N. Arslan for their helpful feedback. Below we respond to the minor comments of A. N. Arslan. The responses (normal font style) are following the *referees' comments* (displayed in italic font style) directly.

*Review comments on tc-2019-142 manuscript, entitled," Towards a webcam-based snow cover monitoring network: methodology and evaluation".*

*Minor Comments:*
*Comment 1: I would recommend adding following references optional which I believe useful to audiences:*

*Page 1: for in-situ measurements:*

*Pirazzini, R.; Leppänen, L.; Picard, G.; Lopez-Moreno, J.I.; Marty, C.; Macelloni, G.; Kontu, A.; Von Lerber, A.; Tanis, C.M.; Schneebeli, M.; De Rosnay, P.; Arslan, A.N. European In-Situ Snow Measurements: Practices and Purposes. Sensors 2018, 18, 2016.*

*Page 21: "In addition, our webcam-based snow cover product can be used to validate Sentinel-2 and Landsat based snow cover products"*

*It would be good to list earlier studies on webcam-based snow cover product which were used for validation satellite-derived products like Sentinel-2 and Landsat. There is one given below*

*Piazzi, G.; Tanis, C.; Kuter, S.; Simsek, B.; Puca, S.; Toniazzo, A.; Takala, M.; Akyürek, Z.; Gabellani, S.; Arslan, A. Cross-Country Assessment of H-SAF Snow Products by Sentinel-2 Imagery Validated against In-Situ Observations and Webcam Photography. Geosciences 2019, 9, 129.*

We included a reference to Piazzi et al. (2019) within the introduction on page two.

the terrain to the webcam, as well as the slope and orientation of the terrain (see Sect. 5 for an in-depth discussion). Webcams may offer detailed analyses of snow cover on steep slopes due to their oblique view on the mountains. Moreover, webcams can provide snow cover information even under cloudy weather conditions as long as cloud cover and fog do not disturb the view on the ground. Therefore, webcams offer an unique potential to complement and evaluate satellite-derived snow information.  For instance, Piazzi et al. (2019) have shown that webcam images can be leveraged to assess the consistency of Sentinel-2 snow cover information. However, the areal coverage of webcam-based snow cover information depends on the number of cameras used, their FOV, and their positioning in the field. In addition, public webcams provide

*Comment 2:*
*Page 3, Line 13: "They propose to apply the blue band classification by Salvatorietal. (2011) and subsequently use principal component analysis (PCA) to separate shaded snow cover from sunlit rock surfaces.."*

*This sentence is not linked to previous one and I believe should be rewritten like what is "they".. NOT CLEAR!*

We corrected the sentence as follows:

[revised manuscript text omitted]